# Influenza Virus Carrying a Codon-Reprogrammed Neuraminidase Gene as a Strategy for Live Attenuated Vaccine

**DOI:** 10.3390/vaccines11020391

**Published:** 2023-02-08

**Authors:** Ji Dong, Zhenyuan Dong, Pei Feng, Yu Gao, Jiashun Li, Yang Wang, Lujie Han, Zhixia Li, Qian Wang, Xuefeng Niu, Chufang Li, Weiqi Pan, Ling Chen

**Affiliations:** 1State Key Laboratory of Respiratory Disease, The First Affiliated Hospital of Guangzhou Medical University, Guangzhou 510182, China; 2State Key Laboratory of Respiratory Disease, Guangzhou Institutes of Biomedicine and Health, Chinese Academy of Sciences, Guangzhou 510535, China; 3Guangzhou nBiomed Ltd., Guangzhou 510535, China

**Keywords:** influenza virus, neuraminidase, codon usage bias, packaging efficiency, live attenuated influenza vaccine

## Abstract

Live attenuated influenza vaccines offer broader and longer-lasting protection in comparison to inactivated influenza vaccines. The neuraminidase (NA) surface glycoprotein of influenza A virus is essential for the release and spread of progeny viral particles from infected cells. In this study, we de novo synthesized the NA gene, in which 62% of codons were synonymously changed based on mammalian codon bias usage. The codon-reprogrammed NA (repNA) gene failed to be packaged into the viral genome, which was achievable with partial restoration of wild-type NA sequence nucleotides at the 3′ and 5′ termini. Among a series of rescued recombinant viruses, we selected 20/13repNA, which contained 20 and 13 nucleotides of wild-type NA at the 3′ and 5′ termini of repNA, respectively, and evaluated its potential as a live attenuated influenza vaccine. The 20/13repNA is highly attenuated in mice, and the calculated LD_50_ was about 10,000-fold higher than that of the wild-type (WT) virus. Intranasal inoculation of the 20/13repNA virus in mice induced viral-specific humoral, cell-mediated, and mucosal immune responses. Mice vaccinated with the 20/13repNA virus were protected from the lethal challenge of both homologous and heterologous viruses. This strategy may provide a new method for the development of live, attenuated influenza vaccines for a better and more rapid response to influenza threats.

## 1. Introduction

Influenza is a highly contagious acute respiratory disease that results in significant morbidity and mortality in humans and domestic animals every year. The disease is caused by the influenza virus, which is an eight-segmented, enveloped RNA virus [1]. Vaccination is considered the most effective method for preventing influenza. Inactivated and live attenuated vaccines are the two main types of influenza vaccines available today [2]. When there is a close antigenic match between vaccine strains and epidemic strains, inactivated vaccines show 70–90% efficacy in reducing the incidence of clinical illness in healthy populations [3,4]. However, inactivated vaccines are limited in generating mucosal immunity and cytotoxic T cell responses and are not sufficiently cross-reactive to protect against antigenic variants. By comparison, live attenuated influenza virus vaccines could induce not only humoral antibodies but also elicit cell-mediated immune responses, which may provide cross protection against antigenic variants [5,6,7].

The advance of plasmid-based reverse genetics techniques makes it possible to engineer the genome of influenza viruses with unique properties that lead to attenuation [8,9]. Many genetically engineered influenza viruses, such as those with deletion modifications in genome segments, have been reported to induce immune responses in animal models [10,11,12,13,14,15,16]. However, some of these vaccine candidate viruses are replication-incompetent and could not be propagated in embryonated chicken eggs or MDCK cells, which are commonly used in the manufacturing system for influenza vaccine.

In this study, we generated replication-competent attenuated recombinant influenza viruses containing codon-reprogrammed NA (repNA) genes by introducing segment-scale silent mutations into the NA gene segment according to mammalian codon usage. The growth potential of these recombinant viruses in embryonated eggs and cells was compared to that of the parental PR8 virus. Furthermore, the attenuation phenotype and the immune response as well as the protective efficacy of the repNA virus were evaluated in a mouse model. By using this codon reprogram strategy, we aim to explore a new approach to developing a live attenuated influenza vaccine candidate strain with properties of convenient production, safety, and efficacy.

## 2. Results

### 2.1. At Least 20 Nucleotides at the 3′ and 13 Nucleotides and the 5′ Termini of Wild-Type NA Coding Regions Are Necessary for Successful Rescue of Recombinant Influenza a Virus Containing Codon-Reprogrammed NA with Impaired NA vRNA Packaging Efficiency

The full coding region of the neuraminidase (NA) gene of influenza virus A/Puerto Rico/8/34 (PR8) was reprogramed according to the *homo sapiens* codon usage bias and then synthesized de novo. The sequence of synthetically reprogrammed NA (repNA) was aligned with the wild-type NA gene. The repNA gene included 283 out of 455 (62.19%) codon changes through 323 out of 1365 (23.67%) nucleotide synonymous substitutions, and the amino acid sequence was completely conserved (Figure 1A). The Codon Adaptation Index (CAI) scores to *homo sapiens* codon usage of the wild-type NA and repNA were 0.73 and 0.95, respectively [17].

Previous studies have shown that both the noncoding region and the coding region at the 3′ and 5′ ends play critical roles in NA vRNA incorporation [18]. To successfully incorporate the repNA, which has multiple silent mutations in the packaging signal region, we restored partial sequences at the 3′ and 5′ ends of the repNA gene to the wild-type NA sequence (Figure 1B). Using eight-plasmid-based influenza reverse genetics, three recombinant influenza viruses were successfully rescued. According to the number of wild-type nucleotides at the 3′ and 5′ ends of the repNA coding region, they were named as 23/42repNA, 20/23repNA, and 20/13repNA viruses, respectively. However, viruses containing full length of repNA and chimeric NA with nucleotides of wild-type PR8 NA less than 20/13repNA, on the other hand, were not rescued. These results suggested that the 20 nucleotides at the 3′ terminus and the 13 nucleotides at the 5′ terminus of the coding sequence are crucial for the incorporation of the NA segment into virions.

We determined the effects of codon reprograming on viral RNA packaging efficiency. The vRNAs were extracted from purified repNA viruses and then analyzed by urea-PAGE and SYBR Green II staining. The band of the NA segment in 23/42repNA was much weaker than that of the wtPR8 virus, while those of 20/13repNA and 20/23repNA were nearly invisible on PAGE gel. However, the RNA bands of the other seven gene segments (PB2, PB1, PA, HA, NP, M, and NS) in each virus were present in approximately equimolar concentrations (Figure 2A). We next determined the quantities of NA vRNA in the viral genome using a relative quantification PCR assay. The results showed that the packaging efficiency of the NA segments of 20/13repNA, 20/23repNA, and 23/42repNA viruses was 0.2%, 2.1%, and 23.1%, respectively, of that of wild-type NA vRNA (Figure 2B). These results suggested that the partial restoration of wild-type NA sequences at the 3′ and 5′ ends of the repNA coding region is necessary for the repNA vRNA to be packaged into the viral genome.

We further explored the effect of codon reprogram manipulation on the protein expression of the NA gene in repNA virus-infected MDCK cells. Compared to the NA segment of wtPR8, the NA protein band of repNA viruses was dramatically weakened. However, the detection of other segments (HA, NP, and M genes) in repNA virus-infected cells was comparable to that of the wtPR8 virus (Figure 2C). Furthermore, the NA activity of the virus-infected cell lysates was measured by the MUNANA assay. When the NA activity of the wtPR8 virus infected cell lysate was set to 100%, the NA activity of the 20/13repNA and 20/23repNA viruses decreased significantly, falling to 0.8% and 7.9% of that of the wtPR8 virus, respectively. Meanwhile, the NA activity of 23/42repNA could reach 70.7% of the wtPR8 virus (Figure 2D). The above results showed that the expression levels of the repNA protein were significantly reduced compared with those of wtPR8 NA, especially in 20/13repNA and 20/23repNA viruses.

### 2.2. repNA Viruses Were Replication Competent but Had Different Growth Phenotypes In Vitro

We first determined the multi-cycle viral growth curves in MDCK cells of repNA viruses to analyze their replicative properties in vitro. The titers of the 23/42repNA virus were less than 10-fold lower than those of the wtPR8 virus. However, the 20/13repNA virus replicated with over a 1000-fold reduction compared to the wtPR8 virus throughout the time course. The growth curve of the 20/23repNA virus was between those of the 20/13repNA virus and the 23/42repNA virus (Figure 3A). We also determined the viral growth of these viruses in chicken embryonated eggs. Similarly, the 20/13repNA virus replicated the worst among all these viruses. Nevertheless, the maximum titer of 20/13repNA at 60 h post-inoculation could reach up to 6.2 log_10_ units (ffu/mL) in allantoic fluids (Figure 3A). The plaques formed by repNA viruses were compared with those of wtPR8 in MDCK cells. The results showed that plaques formed by the 23/42repNA and 20/23repNA viruses were slightly smaller than those formed by the wtPR8 virus. However, the diameters of the plaques formed by the 20/13repNA virus were significantly reduced compared to those of the wtPR8 virus (Figure 3B).

### 2.3. The 20/13repNA Virus Is Highly Attenuated in Mice but Induces Robust Immune Responses

According to the growth properties of repNA viruses in vitro, we selected the most attenuated 20/13repNA virus to determine its pathogenicity in BALB/c mice. Groups of mice were infected with 10^2^–10^6^ ffu of the 20/13repNA virus through intranasal inoculation. The body weight change and survival rate of infected mice were monitored for 14 days. Mice infected with 10^2^–10^4^ ffu of the 20/13repNA virus did not induce any signs of disease but showed slightly delayed weight gain compared to mock-infected mice. Mice receiving 10^5^ ffu of 20/13optNA all survived with just a mild and transient body weight loss. While mice receiving 10^6^ ffu of 20/13repNA showed a significant weight loss, all of them lost more than 25% of their body weight and were humanely killed by 8 dpi. In contrast, mice infected with 10^2^ and 10^3^ ffu of wtPR8 showed significant weight loss starting at 2 dpi, and all mice died by 5~8 dpi (Figure 4A). The calculated LD_50_ value of 20/13repNA (10^5.5^ ffu) was about 10,000-fold higher than that of the wtPR8 virus (10^1.5^ ffu). The viral titers in the lungs of mice infected with 10^3^ ffu of 20/13repNA were over 100,000-fold lower than those of 10^3^ ffu of wtPR8-infected mice on 3 dpi. The viral titers in the lungs of mice infected with 20/13repNA decreased dramatically or cleared on 5 dpi (Figure 4B). These results indicated that the 20/13repNA virus was highly attenuated in vivo.

An important feature of a live, attenuated virus vaccine is that it mimics a natural infection without causing disease but can elicit a potent immune response to confer protection. To evaluate the immune responses induced by 20/13repNA virus, we first investigated the influenza-specific IgG antibody and hemagglutination inhibition (HI) titers in mice intranasally inoculated with 20/13repNA virus. The mice vaccinated with 10^2^–10^4^ ffu of the 20/13repNA virus showed substantial levels of virus-specific serum IgG and HI titers in mice serum at 4 weeks post vaccination. The average HI titers could reach 1:256 in the 10^4^-ffu vaccination group and near 1:64 HI titers in the 10^2^-ffu vaccination group (Figure 5A). One advantage of intranasal immunization with a live attenuated virus vaccine over intramuscular injection of an inactivated virus vaccine is the potential to elicit a mucosal immune response such as specific secreted IgA (sIgA) in the respiratory tract. Therefore, we further determined the virus-specific IgA titers of bronchoalveolar fluid (BALF) and nasal wash samples by ELISA. Mice vaccinated with the 20/13repNA virus induced a significantly higher and dose-dependent mucosal antibody response than mock-vaccinated mice and showed dose dependence (Figure 5A). Another advantage of a live attenuated influenza virus vaccine is its ability to induce a cell-mediated immune response, which may synergize with an antibody response and confer broader cross-reactive protection. To determine whether the 20/13repNA vaccination can induce a cell-mediated immune response, we examined the production of influenza-specific T cells in mice at 10 days after vaccination. Spleen cells from mice were harvested, pooled 10 days post-vaccination, and stimulated with the NP _147–155_ peptide (TYQRTRALV). Mice that received 10^2^–10^4^ ffu of 20/13repNA developed significantly higher numbers of IFNγ-secreting cells than mock-vaccinated mice, which increased with the immunization dose (Figure 5B).

### 2.4. Vaccination with 20/13repNA Protects against Both Homologous and Heterologous Influenza a Virus Challenges

To evaluate the protective efficacy of 20/13repNA virus as a live attenuated vaccine, we challenged 20/13repNA-vaccinated mice with 100 LD_50_ of wtPR8 virus at 4 weeks post vaccination. Mice were monitored for 14 days for body weight loss and survival rate. The lungs’ viral loads were determined 3 days post-challenge. All mice mock vaccinated with PBS showed significant clinical symptoms and died within 5 days post-challenge. Mice receiving 10 ffu of 20/13repNA showed clinical symptoms including ruffled fur, listlessness, and bodyweight loss, and two out of five mice died at 6 days of challenge. Mice that received 10^2^–10^4^ ffu of 20/13repNA showed no clinical symptoms and were completely protected against a lethal challenge, although some 10^2^ ffu-vaccinated mice had temporary and modest weight loss but recovered rapidly (Figure 6A). Virus titers in the lungs of 20/13repNA mice vaccinated with 10-10^3^ ffu were significantly lower than in mock-vaccinated mice and dose-dependent, whereas mice vaccinated with 10^4^ ffu had no detectable viruses (Figure 6B).

We further explored whether 20/13repNA could provide cross protection against heterologous influenza A viruses. At 4 weeks after vaccination, groups of mice vaccinated with 10–10^4^ ffu of 20/13repNA were challenged with 10 LD_50_ of mouse-adapted H1N1 strains (A/California/04/2009, Ca04) or H3N2 strains (A/Aichi/2/1986, X31). In the Ca04 virus challenge test, mice vaccinated with 10^3^ and 10^4^ ffu of 20/13repNA showed no clinical symptoms, while 10^2^ ffu-vaccinated mice had a transient weight loss of less than 20% of body weight and all mice survived (Figure 6C). However, mice vaccinated with 10 ffu of 20/13repNA all died within 6 days post-challenge, the same as those mock-vaccinated mice. In the X31 virus challenge test, mice vaccinated with 10^2^–10^4^ ffu of 20/13repNA were completely protected from lethal challenge, while mice vaccinated with 10 ffu of 20/13repNA and mock-vaccinated mice all died by 8 and 6 days post challenge, respectively (Figure 6C).

### 2.5. 20/13repNA Remains Genome Stable after Multiple Passages in Chicken Eggs

To check the genetic stability of the 20/13repNA virus in vaccine production, the virus underwent nine consecutive passages in embryonated chicken eggs to assess the viral genetic stability. The sequence identities of both reprogrammed codons in the NA and HA genes of passages 1 (P1), 3 (P3), 6 (P6), and 9 (P9) were determined. No mutations were detected in the NA and NA genes of the 20/13repNA virus (Table 1). Meanwhile, the NA vRNA of the 20/13repNA virus remained poorly packaged over the course of nine passages (Table 1). These data indicate a high degree of genetic stability for the 20/13rep NA virus.

## 3. Discussion

Live attenuated influenza vaccine has several advantages over inactivated vaccines, including the elicitation of serum IgG and mucosal IgA antibodies, as well as a T cell response that provides protection at both systemic and mucosal levels. Neuraminidase (NA) is one of the two major surface glycoproteins of the influenza A virus and possesses the enzymatic activity that removes terminal sialic acid residues from glycoconjugates of host cells and hemagglutinin (HA). In previous studies, NA-deficient mutant viruses have been demonstrated to be safe and could produce humoral, mucosal, as well as cell-immune responses that could provide cross-protection for mice and ferrets from lethal virus challenge [19,20]. However, the growth of such influenza viruses lacking NA is dependent on cell lines expressing reduced levels of sialic acid [10] or exogenously added bacterial sialidase [14], which are not approved in vaccine production practice. Although it had been demonstrated that NA-deficient viruses could replicate in the absence of sialidase activity after multiple passages in eggs, the adaptation course was time-consuming, and several mutations appeared around the receptor-binding site of hemagglutinin (HA), and thus the antigenicity of the virus may be changed [21]. Therefore, it is necessary to develop an attenuated but replication-competent influenza virus for efficient vaccine manufacture.

In this study, we reprogramed the coding region of the NA gene (repNA) by introducing full-gene-scale synonymous codon changes based on mammalian codon usage bias (codon reprogram). On the one hand, the packaging efficiency of repNA vRNA is impaired by the multiple silent mutations introduced in the sequence of packaging signals of the NA gene segment. On the other hand, codon reprogram manipulation of repNA was predicted to increase the expression of neuraminidase in mammalian cells. Through the above two aspects of manipulation, we attempted to make an attenuated but replication-competent influenza virus with limited NA vRNA segment packaging into the virions while still providing sufficient neuraminidase activity for virus propagation. Although the 183 and 157 nucleotides at the 3′ and 5′ ends of the NA coding region have been demonstrated to be necessary for peak efficiency of NA vRNA incorporation into the virions [18], we have no idea about the least number of packaging signals needed for NA vRNA incorporation. By gradually restoring the nucleotides at the 3′ and 5′ ends of repNA to wild-type NA, we found that 20 and 13 nucleotides at the 3′ and 5′ ends of the coding region of wild-type NA are necessary to support the limited packaging of codon-reprogrammed NA vRNA segments into the virions. A NA vRNA segment containing different codons may require different lengths of original packaging signals at the 3′ and 5′ ends.

We also noticed that although the packaging efficiency of the NA segment in 20/13repNA virus was only 0.2% of that of the wild type of PR8 virus, it could replicate to over 4–6 log10 ffu/mL in MDCK cells and chicken embryonated eggs, respectively. These results indicated that although the NA activity of the 20/13repNA virus was dramatically reduced, it could still support competent replication without exogenous help. In a mouse model, the 20/13repNA virus showed a highly attenuated phenotype but could induce a dose-dependent humoral and cellular immune response. Mice vaccinated once, even with as low as 10^2^ ffu of 20/13repNA, could be completely protected from both the lethal challenge of homologous and heterologous H1N1 and H3N2 influenza viruses. These results suggested that the highly attenuated 20/13repNA virus could provide a substantial and cross-protective immune response in mice.

Taken together, our findings offer a proof-of-concept strategy for creating a live attenuated influenza vaccine by modifying the NA gene vRNA incorporation and NA protein expression level. It is unlikely that the large-scale synonymous mutations in the NA vRNA will cause the wild-type virulence reversion concern that has been raised by the limited number of amino acid replacements in the traditional live attenuated vaccine. Live influenza vaccines have always been questioned due to possible reassortment between field strains and attenuated vaccine strains during epidemics and pandemics. However, such concerns are unfounded because even if reassortment occurs, the pathogenicity of the resulting reassortment would be the same or less than that of the field strain as long as the backbone virus for the live vaccines is less pathogenic than the field strains. Unfortunately, this NA codon reprogram strategy is unable to overcome the current influenza vaccines’ limitations, including the fact that it cannot offer universal protection against various human influenza virus infections. Further studies are required to explore the possibility of using this NA codon reprogram methodology for other gene segments and segment combinations or in combination with other approaches for the future development of potent vaccine candidates to prevent influenza viral infection.

## 4. Materials and Methods

### 4.1. Ethics Statement

The mouse study protocols (202014 and 202040) were approved by the Institutional Animal Care and Use Committee of the Guangzhou Medical University. The animals were housed and handled in accordance with the guidelines set by the Association for the Assessment and Accreditation of Laboratory Animal Care. All efforts were taken to minimize animal suffering and use the minimum number of animals.

### 4.2. DNA Synthesis and Plasmid Construction

The coding region of the NA gene with altered codon usage bias according to mammalian codon usage was synthesized de novo at Synbio (Suzhou, China). The full-length or partial-length of the wild-type NA sequences in the recombinant pM-PR8NA vector was replaced with the corresponding part of the synthesized codon-reprogrammed NA gene (repNA) by a recombinant reaction using the ClonExpress II One Step Cloning Kit (*Vazyme Biotech Co., Ltd.,* Nanjing, China), following the manufacturer’s instructions. The schematic representations of the chimeric NA vRNA segment of the wild-type and codon-reprogrammed NA genes are shown in Figure 1A.

### 4.3. Virus Rescue

Recombinant viruses possessing codon-reprogrammed NA gene segments were rescued in the gene background of A/Puerto Rico/8/1934 (PR8) as described previously [22]. Briefly, 1 µg of each plasmid (pM-PB2, -PB1, -PA, -HA, -NP, -repNA, -M, and -NS) was transfected into a mixture of 293T and MDCK cells using Lipofectamine 2000 (*Invitrogen,* Carlsbad, CA, USA) according to the manufacturer’s instructions. Forty-eight hours after transfection, culture medium was collected and subsequently inoculated into the allantoic cavities of 10-day-old embryonated chicken eggs for virus propagation. The rescued viruses were named after the construction of a recombinant plasmid containing the repNA gene, such as 23/42repNA, 20/23repNA, etc.

Rescued viruses were propagated in embryonated chicken eggs for 2 days at 37 °C to generate working stocks of the virus. In some experiments, viruses were concentrated and purified by sucrose-gradient ultracentrifugation, as described previously [23].

### 4.4. Growth Curve Analysis

The growth characteristics of viruses carrying the repNA gene were analyzed by infecting MDCK cells and 10-day-old embryonated chicken eggs. Confluent MDCK cell monolayers in a six-well plate were infected with 0.001 multiplicities of virus infection. Infected cells were incubated at 37 °C in DMEM containing 0.3% Bovine Serum Albumin (BSA) (*Invitrogen,* Carlsbad, CA, USA) and 1 µg/mL TPCK-trypsin (*Sigma-Aldrich,* St. Louis, MO, USA). Ten-day-old embryonated chicken eggs were inoculated with 100 ffu of virus. At a given time point, tissue culture supernatant and allantoic fluid were harvested. All the samples were stored at −80 °C until titration by immunostaining the infectious focus forming assay (ffu/mL) on MDCK cells.

### 4.5. Immunostaining Infectious Focus Forming Assay

Viral titers and plaque morphology were determined on confluent monolayers of MDCK cells in six-well plates using a semisolid overlay of 0.8% agarose in minimal Eagle medium (MEM) containing 0.3% BSA and 1 µg/mL TPCK-trypsin. After 72 h of incubation at 37 °C, plaques were visualized by an immunostaining assay, as we described previously [23]. Briefly, the overlays were carefully removed, and the cells were fixed with 4% paraformaldehyde and permeabilized with 0.5% Triton X-100 for 30 min. The cells were incubated for 1 h with an anti-NP monoclonal antibody (*Ambion*, Austin, TX, USA) at a 1:1000 dilution in PBS, followed by 1 h of incubation with peroxidase-conjugated goat anti-mouse IgG (*Boster,* Wuhan, China) at a 1:2000 dilution. The viral antigen was visualized by incubating the cells for 30 minutes at room temperature in an AEC staining kit solution (*Sigma-Aldrich*, St. Louis, MO, USA). Stained plates were scanned, and virus foci were counted. The viral titer, foci forming units/mL (ffu/mL), was determined under a microscope. More than 3–5 adjacent cells stained in one area were considered to form one focal unit.

### 4.6. RNA Electrophoresis

RNA electrophoresis was performed as described previously [23,24]. Briefly, total RNA was extracted from the sucrose gradient purified viruses using the MagMAX Viral RNA Isolation Kit (*Ambion*, Austin, TX, USA). Precipitated RNAs were resuspended in 10 mM Tris buffer (pH 8.0) and then electrophoresed in a 2.8% polyacrylamide gel containing 7 M urea at 80 V for 5 h. RNA molecules were visualized using SYBR Green II RNA Gel Stain (*Invitrogen*, Carlsbad, CA, USA) according to the manufacturer’s protocol.

### 4.7. Real-Time Quantitative PCR

Total RNA was extracted from allantoic fluid containing viruses using the QIAamp Viral RNA Kit (*Qiagen*, Germantown, MD, USA) according to the manufacturer’s instructions. The reverse transcription (RT) reactions were conducted using a universal 3′ primer (5′-AGCA/GAAAGCAGG) and the SuperScript III first-strand synthesis system according to the manufacturer’s protocol (*Invitrogen*, Carlsbad, CA, USA). Ten-fold series-diluted RT products were used as templates for real-time quantitative PCR (qPCR). The qPCR was carried out in a 20 µL ready-to-use SYBR green reaction mixture (*Takara,* Kyoto, Japan) with gene-specific primers for NP (5′-TGTATGGACCTGCCGTAGC-3′ and 5′-CCCTCTTGGGAGCACCTT-3′), optNA (5′-CCCAGGAGTCTGAGTGTGC-3′ and 5’-ACCTTGCCCTTCTCAATCTTG-3′), or wtNA (5′-CAAATGGGACTGTTAAGGACAG-3′ and 5′-TGACCAAGCAACCGATTCAA-3′), or HA (5′-GCATCATCACCTCAAACGCATCA-3′ and 5′-TCAATTTGGCACTCCTGACGTA T-3′) in a Chromo4 Real-Time PCR Detector (*Bio-rad*, Berkeley, CA, USA). The PCR conditions were 95 °C for 1 min and 45 cycles of 95 °C for 5 s, 60 °C for 30 s, and 70 °C for 10 s. The relative concentration of NA vRNA was analyzed by the 2^−ΔΔCT^ method, as previously described [25,26,27]. In brief, the amount of NA vRNA was normalized first by equalizing the level of NP vRNA and then by calculating the percentage of the incorporation of repNA vRNA relative to the level of wild-type PR8NA vRNA. The packaging efficiency of each virus was measured in triplicate using two different virus preparations. The results are presented as the average level of tested vRNA incorporation ± standard deviations.

### 4.8. Western Blot Assay

Confluent MDCK cell monolayers in six-well plates were infected with influenza viruses (2 moi). Twelve hours post-infection, cell lysates were subjected to denaturing SDS-PAGE and blotted onto polyvinyl-difluoride membranes (Millipore) for 1 h. After being blotted, the membrane was incubated for 1 h with the specific rabbit anti-NA hyperimmune serum (*Abcam,* Cambridge, UK), anti-HA (*Sino Bio*, Beijing, China), anti-NP (*Sino Bio,* Beijing, China), anti-M1 (*Novus,* Centennial, CO, USA), and anti-actin (*Biotime,* Beijing, China) antibodies. After being washed, the membrane was incubated for 1 h with HRP-conjugated goat anti-rabbit or goat anti-mouse IgG. Following additional wash steps, blots were detected by chemiluminescence.

### 4.9. Neuraminidase Activity Assay

The NA activity of repNA viruses was measured by a standard fluorometric enzyme assay [28]. MDCK cells were infected with repNA viruses at an MOI of 2. At 12 h post infection, cell lysates were added to the NA fluorogenic substrate, 2’-(4-methylumbelliferyl-N-acetylneuraminic acid (4-MUNANA; *Sigma-Aldrich,* St. Louis, MO, USA), to a final concentration of 100 μM. The reactions were carried out in 50 μL of 33 mM MES (pH 6.5) containing 4 mM CaCl_2_ in 96-well black Opti-plates (*BD Biosciences,* San Jose, CA, USA) and incubated in a 37 °C water bath for 1 h. The reactions were terminated by adding 150 μL of a stop solution containing 0.5 M NaOH (pH 10.7) and 25% ethanol. The fluorescence of released 4-methylumbelliferone was measured using a Labsystems Fluoroskan II spectrophotometer (*PerkinElmer,* Waltham, MA, USA). The excitation wavelength was set at 355 nm, and the emission wavelength was set at 460 nm. Samples were done in triplicate, and the experiments were repeated at least three times.

### 4.10. Animal Experiments

Six-week-old female BALB/c mice, anesthetized with isoflurane, were infected with different doses of 20/13repNA virus and wtPR8virus intranasally (i.n.). The body weight and survival of mice infected with these viruses were monitored for 14 days after inoculation. Control mice were inoculated with 50 µL of PBS only (mock). Mice were humanely killed if they were severely ill and weight loss approached 25%. The 50% lethal dose (LD_50_) values were calculated by the Reed–Muench method [29]. To assess viral replication titers, mice in each group (*n* = 3) were sacrificed at days 3 and 5 after inoculation. The lungs were aseptically removed and homogenized in 1 mL of PBS. The virus titer (ffu/mL) was determined by plaque assay and immunostaining on MDCK cells as described above.

At 4 weeks post vaccination, different groups of immunized mice were challenged intranasally with 100 LD_50_ does of wtPR8 virus, 10 LD_50_ of mouse adapted H1N1 strain (A/California/04/2009, Ca04) or H3N2 strain (A/Aichi/2/1986, X31), respectively. At three days after challenge, the replications of the wtPR8 challenge virus in the lungs of vaccinated mice were titrated in MDCK cells as described above. The remaining animals were observed for body weight and survival for 14 days after the challenge.

### 4.11. Detection of Virus-Specific Antibody

Four weeks after the initial immunization, mice were sacrificed to obtain sera, nasal washes, and trachea-lung washes as described previously [30]. Serum IgG and IgA antibodies in nasal washes and trachea-lung washes were detected by an enzyme-linked immunosorbent assay (ELISA). In the ELISA assay, 96-well polystyrene microtitre plates were coated with purified total wtPR8 virus protein at 4 °C overnight. After incubation of virus-coated plates with serially diluted samples, bound antibodies were detected with horseradish peroxidase (HRP), conjugated secondary goat anti-mouse IgG peroxidase (*Boster*, Wuhan, China), or rabbit anti-mouse IgA (*Southern Biotech*, Birmingham, AL, USA).

Hemagglutinin inhibition (HI) antibody detection was performed according to a WHO-recommended protocol (World Health Organization, 2002). Mice sera were treated with receptor-destroying enzyme (RDE, Denka Seiken) for 16 h at 37 °C prior to heat inactivation for 30 min at 56 °C. The HI assay was The HI titer was the reciprocal of the highest serum dilution that could completely inhibit hemagglutination. The detection limit was a titer of 1:10.

### 4.12. ELISPOT Assay

Spleen cells were collected 10 days after virus inoculation and analyzed for the presence of influenza virus-specific IFN-γ-secreting cells as described previously [31]. In brief, 96-well Multiscreen HA nitrocellulose plates (*Millipore*, Burlington, MA, USA) were coated with 4 µg/mL of rat against mouse IFN-γ monoclonal antibody (R4-6A2, *BD PharMingen*, San Diego, CA, USA) in 50 µL PBS and incubated at 4 °C overnight. Plates were washed and blocked with (RPMI 1640 supplemented with 10% FCS, 50 µM 2-ß-mercaptoethanol, penicillin 100 U/mL, and streptomycin 100 µg/mL) for 2 h at 37 °C. Spleen cells (10^6^ cells/well) from immunized mice were then added to antibody-coated wells. The cells were incubated for 22 h at 37 °C in complete RPMI medium together with 2µg/mL of NP147-155 peptide (NP peptide; TYQRTRALV). After extensive washes, the spots were revealed by successive incubations with biotinylated rat anti-mouse IFN-γ antibodies (XMG1.2, *BD PharMingen*, San Diego, CA, USA), alkaline phosphatase-conjugated streptavidin (*BD PharMingen*, San Diego, CA, USA), and 5-bromo-4-chloro-3-indolylphosphate/nitroblue tetrazolium (BCIP/NBT, *Sigma-Aldrich,* MO, USA) as substrates. Spots in each well were counted using an ELISPOT reader (BioReader 4000, *Bio-Sys GmbH*, Karben, Germany), and the results were expressed as the mean number of IFN-γ secreting cells ± standard deviation (SD) of triplicate mice spleen samples.

### 4.13. Genetic Stability

The 20/13repNA virus was serially passaged in 10-day-old embryonated chicken eggs to assess the genetic stability of the introduced mutations. Eggs were infected with 100 ffu of plaque-purified 20/13repNA virus. Allantoic fluid was collected, and the ffu titer was measured to determine the dilution for subsequent passage of the virus. Subsequent passages were performed in the same manner until passage 9 (P9). For mock-infected controls, PBS was used without virus. At passages 1, 3, 6, and 9, the HA segment (sense 5′-AGCAAAAGCAGGGGAAAATAAAAAC-3′ and antisense 5′-AGTAGAAACAAGGGTGTTTTTCC-3′) and NA segment (sense 5′-AGCGAAAG CAGGGGTTTAAAATG-3′ and antisense 5′-TAGAAACAAGGAGTTTTTTGAAC-3′) were amplified by reverse transcription (RT-) PCR and then sequenced using the Sanger method on an automatic sequencer (Genetic Analyser 3730XL; *Applied Biosystems*, Foster, CA, USA). The packaging efficiency of NA vRNA for indicated passages of 20/13repNA virus was determined by real-time quantitative PCR as described above.

### 4.14. Statistical Analysis

Statistical significance was determined using an unpaired Student’s *t*-test with two-tailed analysis and the *GraphPad Prism* software package Version 5 (*GraphPad* Software, San Diego, CA, USA). Data are considered significant when *p* values are <0.05.

## Figures and Tables

**Figure 1 vaccines-11-00391-f001:**
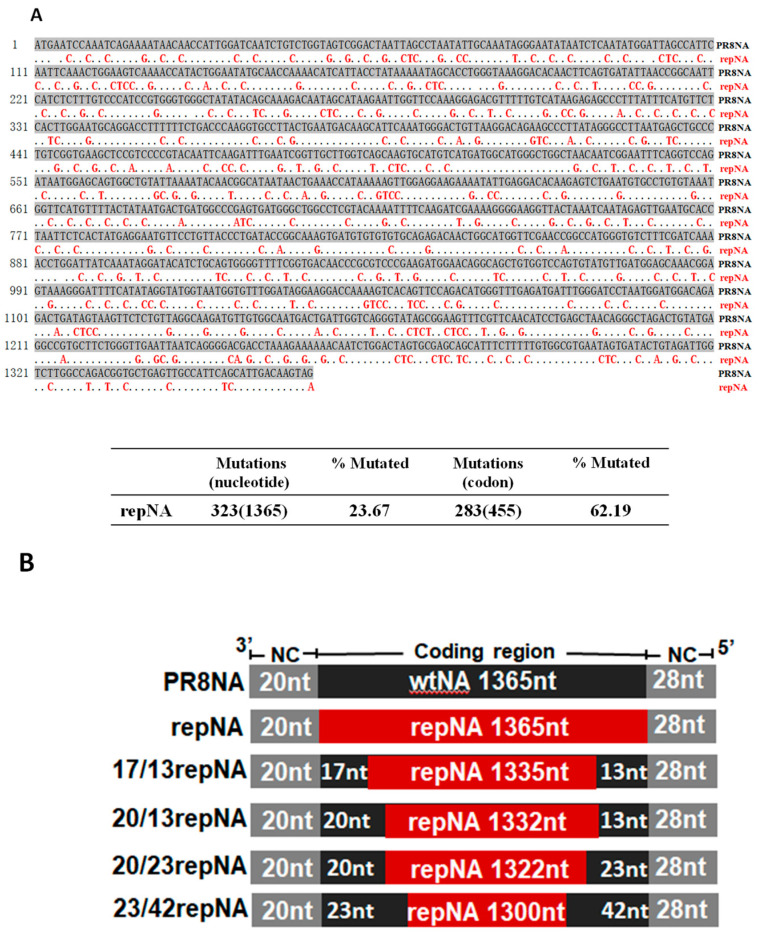
Design of the codon-reprogrammed NA gene. (**A**) Sequence alignment of wild-type and codon reprogramed NA (repNA) genes of PR8 virus (upper panel) and the number and percentage of synonymous nucleotide and codon mutations in the repNA gene (lower panel). The dots in the repNA sequence represent nucleotides identical to those in PR8NA. (**B**) A schematic representation of chimeric wild-type and codon-reprogrammed PR8NA genes. Noncoding regions (NCR) are indicated with gray boxes. The wild-type PR8 NA gene and codon-reprogrammed NA gene are indicated with black boxes and red boxes, respectively.

**Figure 2 vaccines-11-00391-f002:**
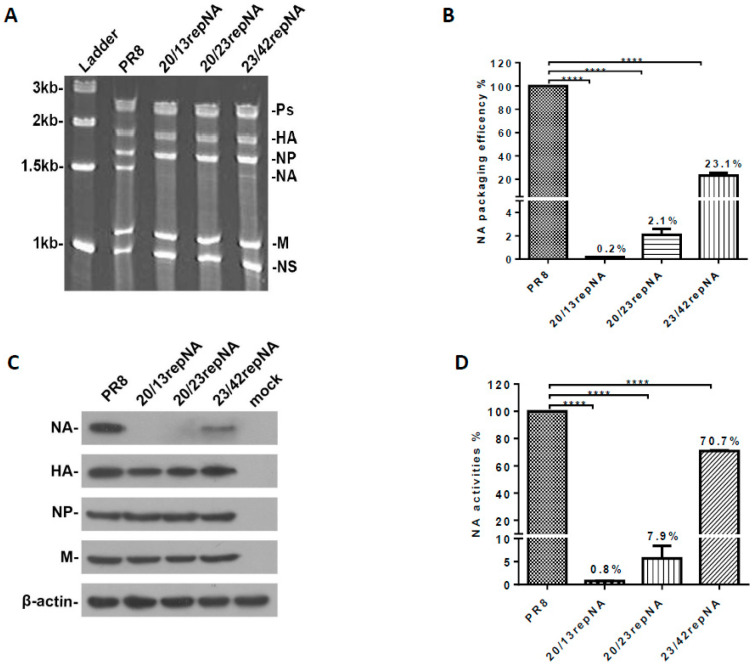
Analysis of the virion packaging efficiency and expression of the codon-reprogrammed NA gene. (**A**) Polyacrylamide gel electrophoresis of viral RNA. vRNAs were extracted from the purified viruses. One microgram of purified viral was loaded onto a 2.8% polyacrylamide gel containing 7.7 M urea and visualized following staining with SYBR Green II. The RNA segment that encodes the polymerase proteins (Ps), hemagglutinin (HA), nucleoprotein (NP), neuraminidase (NA), matrix (M), nonstructural protein (NS), and molecular weight of the ladders are indicated. (**B**) The packaging efficiency of the NA vRNA segment was determined by real-time quantitative PCR. The results are presented as the percentage of incorporated repNA vRNA relative to that of the wild-type PR8 virus. The average values of repNA packaging efficiency from three independent experiments are indicated in the columns. Bars denote standard deviations (SD). (**C**) Western blot analyses of viral protein expression in repNA viruses or wild-type PR8-infected (2 moi) MDCK cells at 12 h post infection. (**D**) The NA activity determination of repNA viruses by NA-specific substrate MUNANA. The average values of NA activity of repNA virus relative to that of wild-type PR8 virus from three independent assays are indicated on the columns. Bars denote SD. **** *p* < 0.0001 (Unpaired two-tailed Student’s *t*-test).

**Figure 3 vaccines-11-00391-f003:**
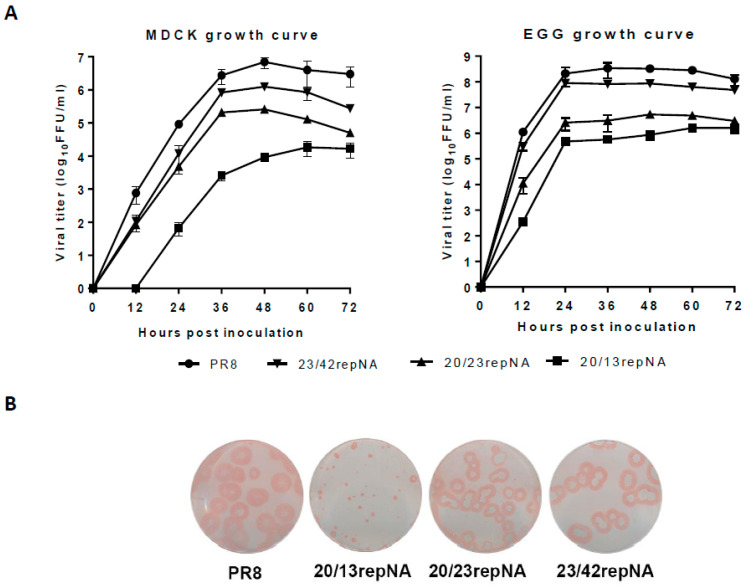
Growth characteristics of repNA viruses in MDCK cells and chicken embryonated eggs. (**A**) MDCK cells were infected with repNA virus or wild-type PR8 virus at an MOI of 0.001 (left panel), and embryonated eggs were infected with 100 ffu of each of the viruses (right panel). At the indicated time after infection, the virus titers in the cell supernatant or allantoic fluid were analyzed by plaque assays. The values are the means and SD from triplicate experiments. (**B**) Plaque phenotypes on MDCK cells of repNA viruses and wild-type PR8 virus.

**Figure 4 vaccines-11-00391-f004:**
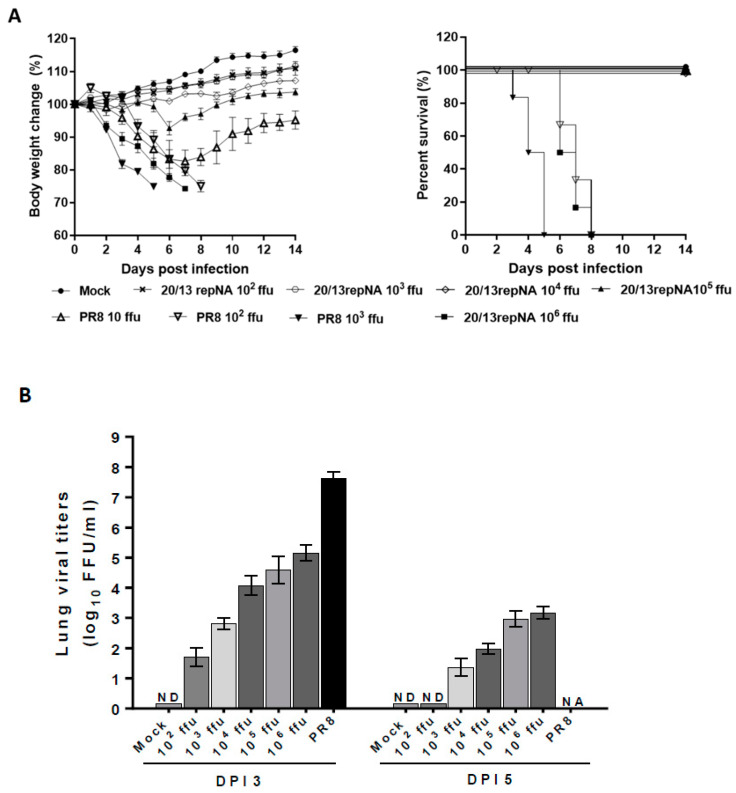
Pathogenicity of 20/13repNA virus in mice. Groups of BALB/c mice were infected intranasally with different doses of the 20/13repNA virus (10^2^~10^6^ ffu) or wild-type PR8 virus (10~10^3^ ffu). Mice treated with PBS were used as controls. (**A**) Body weight (left panel) and survival (right panel) were monitored for 14 days (*n* = 6). Body weight data represent mean ± SEM. (**B**) Lungs were harvested from mice (*n* = 3) at 3 and 5 dpi with different doses of the 20/13repNA virus and 10^3^ ffu of the PR8 virus. Viral titers were determined by plaque assays on MDCK cells. Symbols represent data from individual mice. ND, not detected (detection limit, 10 ffu/mL/lung). NA, not applicable (all mice infected with the PR8 virus died within 5 dpi).

**Figure 5 vaccines-11-00391-f005:**
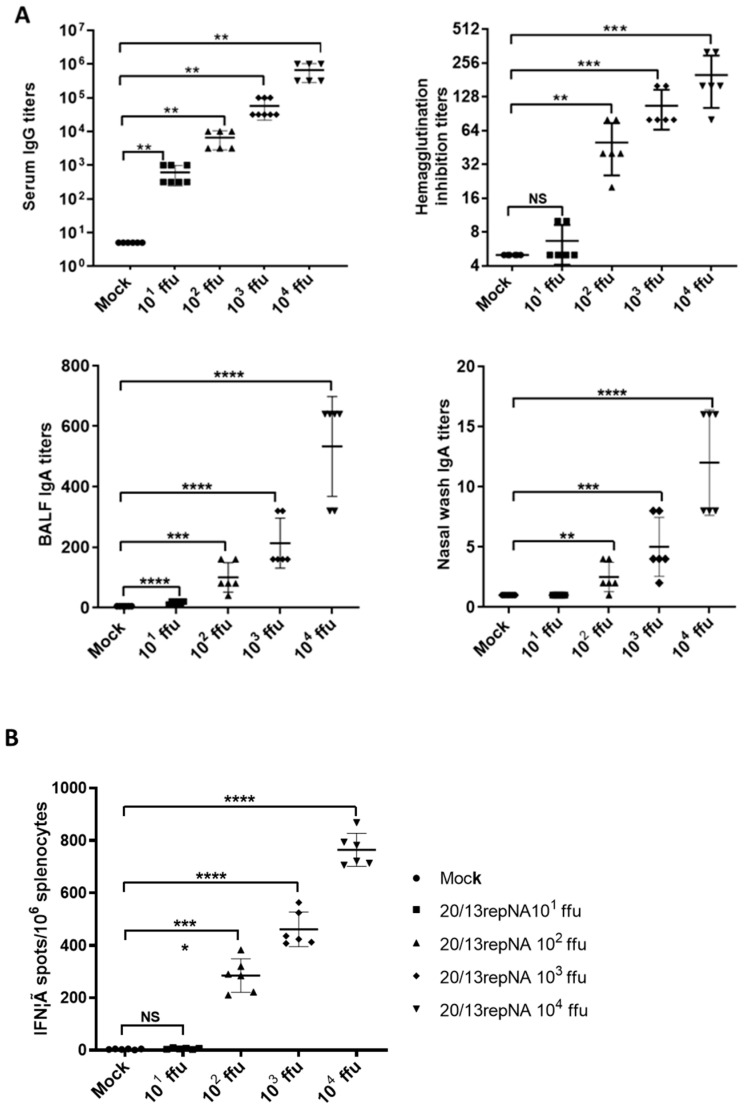
Characterization of the immunogenicity in BALB/c mice. Groups of mice were intranasally vaccinated with different doses of the 20/13repNA virus (10~10^4^ ffu). At 4 weeks post-vaccination, mice (*n* = 6) were sacrificed for serum antibody and mucosal IgA antibody detection. (**A**) Antibody responses, including HI, virus-specific serum IgG, and mucosal IgA antibodies in BALF and nasal washes. Error bars denote SD. The serum samples with an undetectable IgG titer (<10), an undetectable IgA titer in BALF (<10), and an undetectable IgA titer in nasal washes (<2) were given a numeric value of half of the detect limit for statistical analysis. (**B**) Splenocyte anti-IFN-γ producing cell frequency by ELIspot assay. The single-cell suspensions obtained from the spleens of mice (*n* = 6) at 10 days post vaccination were assessed for NP peptide-specific IFN-γ secreting spleen cells. Error bars denote SD. * *p* < 0.05, ** *p* < 0.01, *** *p* < 0.001, **** *p* < 0.0001 (unpaired two-tailed Student’s *t*-test).

**Figure 6 vaccines-11-00391-f006:**
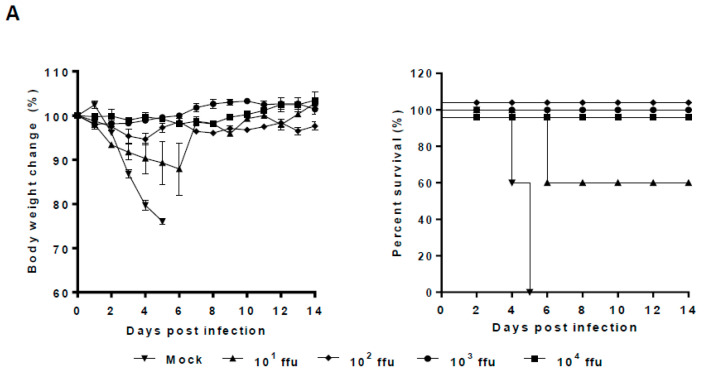
Protective efficacy of 20/13repNA virus against both homologous and heterologous virus challenges. Groups of mice were intranasally vaccinated with different doses of the 20/13repNA virus (10~10^4^ ffu) and then challenged 4 weeks post-vaccination. Body weight change of mice (*n* = 6). (**A**) Virus titers in the lungs of mice (*n* = 3) challenged with 10 LD_50_ of wild-type PR8 virus at 3 days post-challenge. (**B**) Error bars denote SD. ND, not detected (detection limit, 10 pfu/mL/lung). (**C**) Body weight change and survival rate of mice challenged with 10 LD_50_ of Ca04 (H1N1) (upper panel) or X-31 (H3N2) (lower panel). Error bars denote SEM. * *p* < 0.05, *** *p* < 0.001, **** *p* < 0.0001 (unpaired two-tailed Student’s *t*-test).

**Table 1 vaccines-11-00391-t001:** Summary of nucleotide mutations in the HA and NA genes and the packaging efficiency of NA vRNA in the 20/13repNA virus following nine serial passages in chicken embryonated eggs. The dash symbol (-) indicates the absence of specific mutations.

Virus Passage	HA Mutation	NA Mutation	NA vRNA Packaging Efficiency% (Mean ± Standard Errors)
P1	-	-	0.18 ± 0.05
P3	-	-	0.34 ± 0.04
P6	-	-	0.15 ± 0.02
P9	-	-	0.25 ± 0.05

## Data Availability

The materials, data, and any associated protocols that support the findings of this study are available from the corresponding authors upon request.

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
