# Peer review of "Influenza Virus Carrying a Codon-Reprogrammed Neuraminidase Gene as a Strategy for Live Attenuated Vaccine"

_vaccines, 2023, doi:10.3390/vaccines11020391_

Round 1

Reviewer 1 Report

The authors have made a new mutant of the coding region of NA for producing a viable live attenuated vaccine. Although the concept of editing the coding region of NA in influenza virus for vaccine is not new the as was published in this paper prior J Virol. 2004 Mar; 78(6): 3083–3088. Given this information the request from the authors will be to showcase how their paper is different from the previously published literature.

Minor comments:

Figure 4 Figure legend needs to be updated there are only A and B in the figure but A, B, C in the figure legend.

Author Response

Comments and suggestions for authors:The authors have made a new mutant of the coding region of NA for producing a viable live attenuated vaccine. Although the concept of editing the coding region of NA in influenza virus for vaccine is not new the as was published in this paper prior J Virol. 2004 Mar; 78(6): 3083–3088. Given this information the request from the authors will be to showcase how their paper is different from the previously published literature.

Response: Thanks for your comments and suggestions. Although there are several studies have reported that NA-deficient viruses are demonstrated to be safe and could produce overall immune responses that could provide cross-protection for mice and ferrets from lethal virus challenge (J Virol. 2004 Mar; 78(6): 3083–3088; Vaccine, 2005. 23:2922-2927.), these NA-deficient viruses rely on special cell lines expressing reduced levels of sialic acid or exogenously added bacterial sialidase. And these cell lines and bacterial sialidase are not approved in vaccine production practice. In this manuscript, 20/13repNA can replicate competently in common MDCK cells without exogenous help. We have added a few phrases to the discussion to contrast our study with earlier studies. (Page 4~5, Line 193~212).

Minor comments: Figure 4 Figure legend needs to be updated there are only A and B in the figure but A, B, C in the figure legend.

Response: I appreciate your suggestion. The figure legend for figure 4 has been changed. (Page 12, Line 434~435). 

Reviewer 2 Report

The paper, " Influenza virus carrying a codon reprogramed neuraminidase gene as a strategy for live attenuated vaccine " by Ji Dong et al. describes the attenuated recombinant influenza viruses containing codon reprogrammed NA (repNA) gene by introducing segment-scale silent mutations into NA gene segment according to the mammalian codon usage. In my opinion, the work is interesting but needs some revision before published.

Major issues:

1.      Can these reprogrammed NA viruses pass through stably?

2.      Because of the repNA viruses could not express high level of NA protein compared to the WT PR8 virus (Figure 2.), the repNA viruses induced antibody still can recognition virus NA protein or not?

3.      If the packaging efficiency of NA segment in 20/13repNA virus was only 0.2% of that of wild type of PR8 virus, why the authors still codon reprogrammed NA? Or what the difference between 20/13repNA virus and 20/13 WT NA virus?

Minor issues:

1.      The authors could try to use stable expressions NA protein cell line to amplify these repNA viruses to achieve a higher virus titer.

2.      Line 20: “about 10,000-flod higher” should be “about 10,000-fold higher”.

3.      Line 55: “a new approach to develop” should be “a new approach to developing”.

4.      Line 105: “with different growth phonotype in vitro” should be “with different growth phonotypes in vitro”.

5.      Line 114: “in allantonic fluids” should be “in allantoic fluids”.

6.      Line 209: “the expression of neuraminadise” should be “the expression of neuraminidase”.

7.      Line 296: “as described preciously” should be “as described previously”.

8.      Line 335-336: “At 12 h posinfection” should be “At 12 h post infarction”.

9.      Line 408: “virion packaing efficiency” should be “virion packaging efficiency”.

10.   Line 435: “Body weight dada represent” should be “Body weight data represent”.

Author Response

Major issues:

1. Can these reprogrammed NA viruses pass through stably?

Response: Thank you for the suggestion. The genetic stability of the 20/13repNA virus has been determined. The HA and NA genes of the 20/13repNA virus were sequenced using the Sanger method at passages 1, 3, 6 and 9 after the virus had been passaged in chicken embryos for nine passages. At different passages, the 20/13repNA virus's NA gene's packaging efficiency was also assessed. The HA and NA genes of the passaged 20/13repNA virus did not exhibit any alterations. And similar to the original 20/13repNA virus, NA genes from different passages remain poor packaged efficiently. This section of the results is now included in the manuscript as table 1.

2. Because of the repNA viruses could not express high level of NA protein compared to the WT PR8 virus (Figure 2.), the repNA viruses induced antibody still can recognition virus NA protein or not?

Response: I appreciate your suggestion. While the nucleotide of the gene includes synonymous alterations, the amino acid sequence for the repNA gene in our study was totally conserved with the wtPR8 NA protein (Fig.1A). Therefore, both the expression of the NA protein in cells infected with repNA viruses and wtNA viruses could be detected by the specific rabbit anti-NA polyclonal antibody (Abcam). The WB results revealed that the 20/13repNA and 20/23repNA viruses' expression of NA protein was considerably lower than that of wtPR8 (Fig.2C). Additionally, the MUNANA assay was used to gauge the NA activity of the virus-infected cell lysates. The NA activity of the viruses 20/13repNA and 20/23repNA was 0.8% and 7.9%, respectively, of that of the wtPR8 virus. We did not conduct any tests to see if infection with the 20/13repNA and 20/23repNA viruses could sufficiently generate NA-specific antibodies from the minute amount of NA. These antibodies are capable of recognizing virus NA, if they could.

3. If the packaging efficiency of NA segment in 20/13repNA virus was only 0.2% of that of wild type of PR8 virus, why the authors still codon reprogrammed NA? Or what the difference between 20/13repNA virus and 20/13 WT NA virus?

Response: Thank you for the suggestion. In this manuscript, only three recombinant influenza viruses were successfully rescued. They were designated as the 23/42repNA, 20/23repNA, and 20/13repNA viruses, respectively, based on the quantity of wild type nucleotides at the 3' and 5' ends of the repNA coding area. The 20/13repNA virus had the least amount of NA segment protein expression and packaging efficiency among these viruses. Viruses that included the entire repNA and chimeric viruses with wild-type PR8 NA nucleotides that were less than those of 20/13repNA failed to be saved. We also don’t have 20/13 WT NA virus.

Minor issues:

1. The authors could try to use stable expressions NA protein cell line to amplify these repNA viruses to achieve a higher virus titer.

Response: Thanks for your suggestion. It makes sense to assume that repNA viruses could produce a greater virus in steady expression NA protein cell lines. Special cell lines with alterations, however, were not permitted in vaccine manufacture. According to this manuscript, 20/13repNA may competently replicate both in normal MDCK cells and chicken embryonated eggs without the aid of an external source. Additionally, according to Fig. 3A, the highest titer of 20/13repNA at 60 hours after vaccination might be as high as 6.2 log10 units (ffu/ml), which would be sufficient for the generation of live attenuated vaccines.

Minor 2-10:

Response: We appreciate your feedback, and we've fixed the spelling errors in minor suggestions 2 -10.

Reviewer 3 Report

In this study, the authors explored an idea of generating an attenuated influenza A virus by codon reprogramming of its neuraminidase gene. The codon optimization strategy was used to enhance NA protein expression in mammalian cells, while the 3’ and 5’ ends modifications were directed to the alteration of NA RNA packaging into virion, thus making the virus attenuated. The strength of the manuscript is that the authors demonstrated which minimal 3’ and 5’ end sequences are required for packaging of the corresponding RNA, which allow the virus to grow in contemporary substrates, such as eggs and MDCK cells. However, the manuscript has serious limitations that need to be addressed prior to publication.

Major points:

1.            Lanes 221-223: The authors argue that codon optimization of the NA gene resulted in 4-fold more efficient expression of the NA protein compared to the NA RNA packaging, however, this argument is not supported by the data presented on Figure 2. There was no NA band seen on the WB figure for the 20/13 repNA virus. It seems that the main effect of virus attenuation comes from the modified 3’ and 5’ end, and this effect could the same, regardless of the codon optimization of the other NA gene coding part. In the other words, an important control virus is missing in this experiment: the one that has modified 3’ and 5’ packaging signals and the native NA gene of the remaining NA coding fragment.

2.            The authors do not provide data on genetic stability of the NA modified viruses. It is possible that the viruses will rapidly adapt by acquiring mutations in HA protein, as it was the case for the NA deficient viruses. At least 10 sequential passages of the rescued viruses should be performed, followed by full-genome sequencing, to prove that the viruses will maintain their attenuated characteristics.

3.            The authors do not discuss major limitations of their LAIV strategy. Thus, LAIV viruses are updated almost annually and the use of such cumbersome NA modification strategy would not allow rapid generation of the LAIV strains with established safety and immunogenicity profiles. Moreover, this paper deals with PR8 H1N1 virus, but the seasonal LAIVs are prepared from currently circulating H1N1 and H3N2 viruses, which will require additional experimentations to establish necessary 3’ and 5’ packaging signals modifications to achieve the desired phenotypic properties. Another limitation is the possibility of the NA-modified LAIV virus to reassort with wild-type virus, if a vaccinated individual gets infected with any seasonal influenza virus. In this case, the vaccine virus will revert to virulent phenotype.

4.            The authors argue that the 20/13 repNA virus is 10000-fold more attenuated than the parental PR8 virus. However, in their experiments the LD50 for the PR8 strain was not demonstrated, although the authors refer to a number 33 ffu (lane 131). These data are in contrast to the studies by other authors, where the LD50 for the PR8 virus ranged from 2.5 to 3,5 lg EID50. Here, the authors used one dose of 3.0 lg ffu for the PR8 virus, where all mice succumbed to infection. These data may indicate that the LD50 can be as high as 2.5 lg (or 316 ffu), which is significantly different from 33 ffu. The authors should be careful in interpreting the data they generated in this particular study.

5.            The Methods section lacks the description of the challenge experiments.

Minor points:

1.            The structure of the paper doesn’t match the Vaccines template. Please follow the Journal’s instructions for the paper layout.

2.            Lane 105. Replace “phonotype” with “phenotype”.

3.            Lane 139, . Replace “hemagglutinin” with “hemagglutination”

4.            Lane 156. Replace “received with” with either “received” or “immunized with”.

5.            Lanes 226-332. More details are needed to describe how the WB were set up and developed, given that the secondary antibodies were HRP-conjugated. Why chemiluminescence was used here? Which device was used for data reading?

6.            Lanes 356-369. The description of ELISA is misleading. It should be clearly described that IgG antibodies were assessed in serum samples, while IgA antibodies were detected in NW and BAL samples. Moreover, different samples were diluted differently, as can be seen from the data in Figure 5.

7.            The description of ELISpot assay should include the number of cells added to each well.

8.            Figures 3 and 5. There can’t be “0” values. Each assay has some limit of detection.

Author Response

Major points:

1. Lanes 221-223: The authors argue that codon optimization of the NA gene resulted in 4-fold more efficient expression of the NA protein compared to the NA RNA packaging, however, this argument is not supported by the data presented on Figure 2. There was no NA band seen on the WB figure for the 20/13 repNA virus. It seems that the main effect of virus attenuation comes from the modified 3’ and 5’ end, and this effect could the same, regardless of the codon optimization of the other NA gene coding part. In the other words, an important control virus is missing in this experiment: the one that has modified 3’ and 5’ packaging signals and the native NA gene of the remaining NA coding fragment.

Response: We partially agree with your comment that the main effect of virus attenuation comes from the modified packaging signal sequences at 3’ and 5’ end. And the “The codon reprogramed NA (repNA) gene failed to be packaged into viral genome, which was achievable with partial restoration of wild-type NA sequences nucleotides at the 3’ and 5’ termini” is the key point of this manuscript. We postulated the role of codon optimization played in elevating the expression of the deficient packaging of NA segment only in the discussion (Line 208~209). Since the WB result was unable to precisely quantify the expression of the trace protein, we made a roughly calculation based on the NA activity of repNA viruses compared to that of wtPR8 virus (Line 221-222).

The control virus, you mentioned in your comments, should have very poor packaging efficiency as that of 20/13repNA virus. And that means a considerable portion of the NA coding fragment needs to be changed. And the expression of NA is likely to be impacted by these modulations. As a result, we believe it will be challenging to discover the ideal control virus.

2. The authors do not provide data on genetic stability of the NA modified viruses. It is possible that the viruses will rapidly adapt by acquiring mutations in HA protein, as it was the case for the NA deficient viruses. At least 10 sequential passages of the rescued viruses should be performed, followed by full-genome sequencing, to prove that the viruses will maintain their attenuated characteristics.

Response: Thank you for the suggestion. The genetic stability of the 20/13repNA virus has been determined. The HA and NA genes of the 20/13repNA virus were sequenced using the Sanger method at passages 1, 3, 6 and 9 after the virus had been passaged in chicken embryos for nine passages. At different passages, the 20/13repNA virus's NA gene's packaging efficiency was also assessed. The HA and NA genes of the passaged 20/13repNA virus did not exhibit any alterations. And similar to the original 20/13repNA virus, NA genes from different passages remain poor packaged efficiently. This section of the Result is now included in the manuscript as table 1 (Line 186~194ï¼›409-421ï¼›498-505).

3. The authors do not discuss major limitations of their LAIV strategy. Thus, LAIV viruses are updated almost annually and the use of such cumbersome NA modification strategy would not allow rapid generation of the LAIV strains with established safety and immunogenicity profiles. Moreover, this paper deals with PR8 H1N1 virus, but the seasonal LAIVs are prepared from currently circulating H1N1 and H3N2 viruses, which will require additional experimentations to establish necessary 3’ and 5’ packaging signals modifications to achieve the desired phenotypic properties. Another limitation is the possibility of the NA-modified LAIV virus to reassort with wild-type virus, if a vaccinated individual gets infected with any seasonal influenza virus. In this case, the vaccine virus will revert to virulent phenotype.

Response: We appreciate your suggestions. We have added the limitations of repNA virus approach in Discussion section (Line 243~258).

4. The authors argue that the 20/13 repNA virus is 10000-fold more attenuated than the parental PR8 virus. However, in their experiments the LD50 for the PR8 strain was not demonstrated, although the authors refer to a number 33 ffu (lane 131). These data are in contrast to the studies by other authors, where the LD50 for the PR8 virus ranged from 2.5 to 3,5 lg EID50. Here, the authors used one dose of 3.0 lg ffu for the PR8 virus, where all mice succumbed to infection. These data may indicate that the LD50 can be as high as 2.5 lg (or 316 ffu), which is significantly different from 33 ffu. The authors should be careful in interpreting the data they generated in this particular study.

Response: Before a new batch of virus or new mouse strains is utilized, we will determine the lethality of the PR8 virus in animal. Depending on the various viral batches, mouse strains, mouse source, operators, etc., the LD50 of PR8 ranges from 101.5 (33) pfu to 100 pfu in our lab. The LD50 of PR8 in this study was determined as 101.5 (33) pfu, also as the 101.5 ffu(ffu is the immunostaining of pfu by use of ait-NP antibody). We have added the mice bodyweight change and survival rate after infection with different doses of PR8 virus to Figure.4A (Line 460) for calculating the LD50 of PR8. Additionally, we observed that the LD50 of PR8 published in several articles was not uniform; some of them were close to our findings, such as 25 pfu (J Virol. 2005 Mar;79(5):2910-9.), 60 pfu (Nat Biotechnol. 2010 Jul;28(7):723-6.), <100 pfu (Proc Natl Acad Sci U S A. 2010 Jun 22;107(25):11531-6.). While some of them were substantially greater than our LD50 of PR8, including 1000 PFU (PNAS, 2000 April 11; 97(8): 4309–4314), and 2.5 to 3,5 lg EID50 in your comments. The reason for the difference in LD50 of PR8 virus might due to the different strains of PR8 preserved in different labs.

5. The Methods section lacks the description of the challenge experiments.

Response: We appreciate your suggestions. We have added the method of challenge experiments in Method section (Line 370~375).

Minor points (1-8):

Responses:  We appreciate all the minor suggestions. And the corresponding corrections were made, with yellow highlights.

Round 2

Reviewer 3 Report

In the revised version of the article, the authors took into account the main comments of the reviewers and corrected most of the flaws. In general, the manuscript can be accepted for publication, but a couple of comments remain. Still, the sentence "We also noticed that the NA protein expression of the impaired NA vRNA in 20/13repNA virus increased about 4-folds than its NA vRNA incorporation efficiency, which might be resulted from the codon optimization manipulation" on lines 230-232 sounds incorrect, because the efficiency of viral RNA incorporation is assessed by analysis of purified viral particles, whereas the functional activity of neuraminidase is assessed in cells infected with the virus. Therefore, it is incorrect to claim a 4-fold increase in NA protein expression compared with viral RNA incorporation based on the data that 0.2% of viral RNA from control virus is incorporated, while in cells NA activity is 0.8% of the control.

Another minor issue is the suggestion to add the "20/13repNA" virus name on figures 4-6 legends, rather than just "ffu". It is especially important where the WT PR8 control virus is included.  

Author Response

Comments and Suggestions for Authors In the revised version of the article, the authors took into account the main comments of the reviewers and corrected most of the flaws. In general, the manuscript can be accepted for publication, but a couple of comments remain. Still, the sentence "We also noticed that the NA protein expression of the impaired NA vRNA in 20/13repNA virus increased about 4-folds than its NA vRNA incorporation efficiency, which might be resulted from the codon optimization manipulation" on lines 230-232 sounds incorrect, because the efficiency of viral RNA incorporation is assessed by analysis of purified viral particles, whereas the functional activity of neuraminidase is assessed in cells infected with the virus. Therefore, it is incorrect to claim a 4-fold increase in NA protein expression compared with viral RNA incorporation based on the data that 0.2% of viral RNA from control virus is incorporated, while in cells NA activity is 0.8% of the control. Response: Thanks for your suggestion. We have deleted the sentence "the NA protein expression of the impaired NA vRNA in 20/13repNA virus increased about 4-folds than its NA vRNA incorporation efficiency, which might be resulted from the codon optimization manipulation" on lines 230-232.   Another minor issue is the suggestion to add the "20/13repNA" virus name on figures 4-6 legends, rather than just "ffu". It is especially important where the WT PR8 control virus is included.   Response: Thanks for your suggestion. The name of the virus "20/13repNA" has been added to Fig. 4A. We don't believe it is necessary to include this lengthy name in Fig 4B, Fig 5, or Fig 6, as these figures just contain the 20/13repNA virus.